# Simultaneous Observation of Mouse Cortical and Hippocampal Neural Dynamics under Anesthesia through a Cranial Microprism Window

**DOI:** 10.3390/bios12080567

**Published:** 2022-07-26

**Authors:** Rujin Zhang, Chaowei Zhuang, Zilin Wang, Guihua Xiao, Kunsha Chen, Hao Li, Li Tong, Weidong Mi, Hao Xie, Jiangbei Cao

**Affiliations:** 1Department of Anesthesiology, The First Medical Center, Chinese PLA General Hospital, Beijing 100853, China; auau93@163.com (R.Z.); wangzilin1004@163.com (Z.W.); chenkunsha@outlook.com (K.C.); lihao301@126.com (H.L.); tongli301@aliyun.com (L.T.); wwdd1962@aliyun.com (W.M.); 2Department of Automation, Tsinghua University, Beijing 100084, China; zhuangcw16@mails.tsinghua.edu.cn (C.Z.); xiaoguihua@mail.tsinghua.edu.cn (G.X.)

**Keywords:** fluorescence microscope, in vivo imaging, deep imaging, microprism

## Abstract

The fluorescence microscope has been widely used to explore dynamic processes in vivo in mouse brains, with advantages of a large field-of-view and high spatiotemporal resolution. However, owing to background light and tissue scattering, the single-photon wide-field microscope fails to record dynamic neural activities in the deep brain. To achieve simultaneous imaging of deep-brain regions and the superficial cortex, we combined the extended-field-of-view microscopy previously proposed with a novel prism-based cranial window to provide a longitudinal view. As well as a right-angle microprism for imaging above 1 mm, we also designed a new rectangular-trapezoidal microprism cranial window to extend the depth of observation to 1.5 mm and to reduce brain injury. We validated our method with structural imaging of microglia cells in the superficial cortex and deep-brain regions. We also recorded neuronal activity from the mouse brains in awake and anesthesitized states. The results highlight the great potential of our methods for simultaneous dynamic imaging in the superficial and deep layers of mouse brains.

## 1. Introduction

The fluorescence microscope is a powerful tool for mouse brain imaging with high resolution and specific labeling [1]. In in vivo mouse brain imaging, an optical microscope is commonly applied in either a cortex-wide field-of-view (FOV) in the superficial cortex [2] or deep cortex regions with a small FOV [3]. However, the mouse cerebral cortex is formed vertically into six layers ranging from 0 to 700 μm and horizontally into different brain regions, with functional connections among different layers and regions. For example: the primary visual cortex is highly coherent, with specific zones in both superficial and deep laminar domains in the visual cortex [4,5]; chronic epilepsy leads to distinct oscillations in superficial and deep entorhinal cortex when presubiculum stimulated [6]; general anesthesia has been demonstrated to affect the visual cortex [7], the somatosensory cortex [8] and the auditory cortex [9]. In addition, many cognitive processes and diseases in mouse brains involve not only the cortex, but also the deep cerebellar nuclei. For example, recalling of remote contextual fear memory involves interactions between the anterior cingulate cortex and CA1 in the hippocampus. Prefrontal cortex communication with the mediodorsal and ventromedial thalamus is involved in goal-directed behaviors [10]. Early Alzheimer’s disease is associated with widespread Aβ-mediated damage to the entorhinal-hippocampal circuit [11]. Post-traumatic stress disorder (PTSD) is associated with amygdala, medial prefrontal cortex, and hippocampus functioning [12]. To explore these issues, we require microscopes to have brain-wide FOV, video-rate acquisition, and multi-depth imaging.

A variety of techniques have been proposed to solve the problem. Mesoscopes have the advantages of a large FOV and cellular resolution: the firefly microscope [13] and the real-time, ultra-large-scale, high-resolution (RUSH) macroscope [12] have more than 6-mm FOV, cellular resolution and video-rate acquisition, which is applicable to in vivo superficial cortex imaging in mouse brains. However, severe background light and tissue scattering problems prevent them from imaging the deep cortex. Although two-photon mesoscopes have been proposed for cortex-wide imaging in the deep brain, they fail to achieve dynamic acquisition due to their slow frame rate. The two-photon random access mesoscope (2p-RAM) [14] provides a 1.9 Hz frame-rate for 4.4-mm FOV, while the Diesel2p mesoscope [15] achieves 5-mm FOV with a maximum 3.84 Hz frame rate. To further improve imaging depth, adaptive optics (AO) correct the wavefront distortion caused by tissue scattering [16,17], and improve the depth of the two-photon microscope to 760 μm [18]; however, the depth is not sufficient to reach the hippocampus. Benefitting from longer wavelength and higher-order non-linear effects, the three-photon microscope can achieve neuronal imaging in the mouse hippocampus [19,20]. However, it is more difficult to balance the FOV and frame rate as a result of the limited pulse frequency of the laser. Optical fiber photometry [21] and gradient index (GRIN) lenses [22,23] show their superiority in deep brain imaging, penetrating several millimeters in depth. However, fiber photometry fails to provide spatial resolution. Although the GRIN-lens-based endomicroscope achieves real-time, deep multi-region imaging [24], simultaneous observation of cortical neuronal activity is still challenging due to its limited DOF.

A right-angle microprism window chronically implanted into the brain enables simultaneous imaging of multiple cortical layers, including layer V, and has previously been used in other optical microscopy modalities [25,26,27]. Furthermore, the microprism implantation has been shown to cause manageable damage and to be effective for long-term application in mouse brains. However, simultaneous imaging of the superficial cortex and entire cortical columns has not been reported. In addition the chronic microprism window only covers cortical layers, but fails to further image the hippocampus. To extend the imaging depth to the hippocampus, the volume of the microprism inevitably increases, leading to more damage in mouse brains.

In this study, we propose a novel method, which combines wide-field microscopy and chronic microprism windows, to simultaneously observe the whole superficial cortex and deep brain regions. We first implanted the microprism window into mouse brains after craniotomy to improve the imaging depth to 1 mm. Then, we utilized a multi-planar imaging strategy to compensate for the depth difference between the superficial cortex and deep tissues, achieving simultaneous imaging of the superficial cortex and deep longitudinal tissue. To further improve the imaging depth to 1.5 mm with minimized brain damage, we designed a rectangular trapezoidal microprism, providing an imaging depth from 0.5 to 1.5 mm. The results show that our technique was able to image the cellular structures of microglia in the superficial cortex and deep tissues. Furthermore, we performed simultaneous imaging of dynamic neuronal activities in the superficial cortex and the tissue at different depths in the process of anesthesia and recovery from isoflurane and propofol.

## 2. Materials and Methods

### 2.1. Strategy of Multi-Planar Imaging

A schematic diagram of simultaneous multi-planar brain imaging is shown in Figure 1a,b. Firstly, after removing the skull of the mice, we made an incision in the mouse brains with approximately 1-mm width and 1.5-mm depth, and inserted a cranial window (1 mm × 1 mm × 1 mm microprism or 1 mm × 1 mm × 1.5 mm rectangular-trapezoidal microprism) into the brain. The right-angle microprism provides the imaging depth from 0 to 1 mm, and the rectangular trapezoidal microprism provides the imaging depth from 0.5 to 1.5mm. The slant surface of the microprism reflects the longitudinal cortex tissue to the horizontal plane, and the image of the longitudinal tissue is Δ*D* below the superficial cortex. For simultaneous imaging of the longitudinal plane and superficial cortex, we placed a refractive medium on the coverslip to shift the longitudinal plane to the superficial cortex. Based on Snell’s Law, we derived the relationship between the distance difference Δ*D* and the thickness of the refractive media *D_r_* as [28] *D_r_* = Δ*D_n_o__*/(*n_o_* − *n_i_*), where *n_i_* is the refractive index of the imaging plane and *n_o_* is the refractive index of the air. Assuming a dry objective is used for detection at a wavelength around 515 nm, we estimated the refractive indices as *n_i_* = 1 and *n_o_* = 1.5 for glass. The Δ*D* was 1 mm for the right-angle microprism and 1.5 mm for the rectangular-trapezoidal microprism (Figure 1c), so we placed an additional glass column with thickness 2 mm and 3 mm for each case.

### 2.2. Optical Mesoscope Setup

To demonstrate the effect of our method, we built a wide-field mesoscope constructed from off-the-shelf components as described in our previous work (Figure 1d). Briefly, the excitation source is a CW laser (MBL-III-473-100 mW, CNI) at a central wavelength of 473 nm. The laser beam is expanded to 12 mm by a beam expander (BE) and a pair of 4f-system lenses with Lens1 (f = 150 mm) and Lens2 (f = 200 mm). After being focused by Lens3 (f = 150 mm) and being reflected by a microprism, the beam passes through an excitation objective with 2×/0.5 NA (MVPLAPO 2 XC, Olympus) and excites the fluorophores in samples. The microprism placed between the objective lens and the camera plays a similar role as a dichroic mirror, which can separate the coherent excitation laser and the incoherent emission fluorescence [13]. The fluorescence is collected by an epi-fluorescence setup including the same objective, a tube lens (MVPLAPO 1X, Olympus), a filter (wavelength: 520 ± 12.5 nm/630 ± 46.5 nm, 87-241, Edmund) and an sCMOS (ORCA-Flash4.0 V3, HAMAMATSU). The FOV of the system is 6.6 mm and each pixel in the sCMOS corresponds to 3.25 μm on the image plane with 2× magnification and 6.5 μm pixel size. We used a 2-mm right angle aluminum-coated microprism in our system to maximize light efficiency.

### 2.3. Animal Surgery

Chronic craniotomy was performed according to the procedures described in a previous report in accordance with the guidance of the Animal Care and Use Committee of Tsinghua University. Briefly, a window of 7 mm diameter was created. We then made an incision in the mouse brains of about 1-mm width and 1-mm depth with a 0.9 mm diameter (19 gauge) blunt needle. Then the microprism was inserted. To assist the alignment of the prism with the cortex, the vertical face of the microprism was lined up in parallel to the handles on the forceps. A glass coverslip was used to cover the cortex. Lastly, we mounted a head post to the skull and fixed it with dental cement. During the whole surgery procedure, the mouse was anesthetized using isoflurane (3% in oxygen for induction and 1.5–2% for surgery and imaging) and a breathing frequency of 1 Hz was maintained. The body temperature was maintained at 37.5 °C with a feedback-controlled blanket and eye ointment was applied.

### 2.4. Fluorescent Beads Preparation and Anesthesia Treatment

To measure the resolution of our system, 3.1-μm-diameter green fluorescent beads (T14792, Thermo Fisher Scientific) were fixed in a pool with 1% low-gelling temperature agarose (A9414-25G, Sigma, Burlington, MA, USA), and then a microprism was inserted in the pool and covered with a coverslip. In our in vivo mouse experiments, we applied two different anesthesia treatments to the mice. For isoflurane inhalation, the mice were fixed under the microscope and inhaled oxygen through a mask connected to an anesthesia machine (R530IE, RWD life science company, Shenzhen, China) at a flow rate of 2 L/min. At 120 s after the beginning of the experiment, we delivered 2% isoflurane (RWD life science company, China) at an oxygen delivery rate of 2 L/min. At 240 s, we stopped the isoflurane inhalation and waited for the recovery of the mouse. For propofol injection (Fresenius Kabi Austria GmbH company, Graz, Austria), we also placed the mouse under the microscope and provided inhaled oxygen to the mouse. At 120 s after the beginning of the experiment, we injected propofol through the tail vein (26 mg/kg).

## 3. Results

### 3.1. System Characterization

To evaluate the performance of our methods, we first measured the lateral intensity profiles of the fluorescent beads at different depths, as shown in (Figure 2). The 3.1-μm-diameter green fluorescent beads were fixed in a pool with 1% agarose; we then inserted a microprism in the pool and covered it with a coverslip. A glass column was placed above the microprism to simultaneously image the superficial plane (Plane 1, as shown in (Figure 2a) and the longitudinal plane (Plane 2, as shown in (Figure 2a). We imaged the microspheres with a high-performance microscope and measured the lateral intensity profiles of Plane 1 (upper Figure in (Figure 2b)) and Plane 2 (lower Figure in (Figure 2b)). After Gaussian fitting, the full widths at half maximum of the fluorescent microspheres on the two planes were 4.1 μm and 5.0 μm, respectively, indicating that cellular resolution could be obtained in both planes. Because Plane 2 was reflected by a microprism to the camera, all points at different depth in Plane 2 were mapped into the focal plane and had an equivalent optical path so that they should have no difference in resolution.

We performed simultaneous imaging of the superficial cortex and deep longitudinal cortex tissue with high spatiotemporal resolution. By inserting the microprism and a glass column, we were able to observe the neuronal activity at 0–1.5 mm depth. We have described the effects of the optical aberrations, reflection and vignetting introduced by a refractive medium in the collection path in our previous work [28]. However, the microprism can easily be affected by brain tissue during surgery, so if the upper surface of the microprism is not perpendicular to the optical axis, it causes additional aberrations. To illustrate the effect of the aberrations, we simulated the point-spread-function (PSF) with all parameters the same as in our system. In Figure 3c, the microprism is completely perpendicular to the optical axis with a 3-mm thickness glass column and the NA of the objective is 0.5. The result shows the 0.76 μm lateral resolution in this case. When we changed the tilt angle of the glass surface, the PSF was degraded. When we set the tilt angles to be 5° and 9° in our simulation (Figure 3d,e), the lateral resolutions were 0.79 μm and 0.82 μm, respectively, showing the resolution was slightly affected by the tilt angle. However, when the tilt angle was increased to 10° (Figure 3d,f), the lateral resolution became 3.00 μm, indicating that the lateral resolution was degraded severely by more than 10°.

### 3.2. In Vivo Imaging of Microglial Cells in Mouse Brains by Right-Angle Microprisms

We then performed our proposed technique for in vivo imaging of microglia cells in Cx3Cr1-GFP mice (JAX No. 005582). The right-angle microprism was applied to obtain a longitudinal view with 0–1000 μm depth. Figure 4a shows that the cellular structures of the microglia cells across the superficial cortex and in the longitudinal plane were captured by the wide-field mesoscope in a single snapshot. To better show the details, we selected two subregions of the longitudinal plane (marked in the red box) and the superficial cortex (marked in the yellow box) and zoomed in Figure 4b,c, respectively. The microglia cells on the longitudinal plane and in the superficial cortex were distinguished clearly and had similar structural details, showing that our method achieved simultaneous imaging of the superficial cortex and the deep cortex in the mouse brains. As shown in Figure 4b, the microglia cells were distributed in the depth range of 0–1000 μm, which corresponds to different horizontal positions in the image.

### 3.3. Deep In Vivo Functional Imaging of Neuron Cells in Mouse Brains by Rectangular Trapezoidal Microprism during Anesthesia

Finally, we demonstrated deep real-time functional imaging of the neurons in Ai148 (TIT2L-GC6f-ICL-tTA2)-D;Rasgrf2-2A-dCre (JAX 030328, JAX 022864) mouse brains. Adeno-associated virus (AAV) (rAAV-hSyn-GCaMp6f-WPRE-hGH pA, 200 nL; 1.03 × 10^12^ μg/mL, BrainVTA, Wuhan, China) was injected into the medial parietal association cortex (MPtA, ML: −1.3 mm, AP: −2.0 mm, DV: −1.0 mm) of the mice for deep brain imaging. Although the right-angle microprism provided a longitudinal view of the deep mouse tissue, we could still not observe the neuronal activity in the hippocampus. Thus, a rectangular trapezoidal microprism was applied to provide the longitudinal view with depth at 500–1500 μm, covering the deep cortex and hippocampus. By this method, we recorded the neuronal activity in the superficial cortex and hippocampus simultaneously in mouse brains in awake and anesthesitized states with a 10 Hz frame rate.

To observe the neuronal activity in mouse brains anesthetized by isoflurane inhalation (RWD life science company, Shenzhen, China), we took 8000 time-lapse images with a 10 Hz frame-rate (800 s in total acquisition time). The first 1200 time-lapse images (120 s) were captured for awake mice brains; the standard deviation projections are shown in Figure 5a. Then, 2% isoflurane at a 2 L/min delivery rate of oxygen was delivered to the mice via a gas mask, as shown in Figure 5b. After 120 s, we stopped the isoflurane anesthesia and recorded the neuronal activity of the recovery. Two subregions of the superficial cortex and longitudinal plane labeled in Figure 5a were zoomed in on and are shown in Figure 5c,e; the neurons were visualized in both the superficial cortex and the longitudinal plane. We selected several neurons in the superficial cortex and the deep tissue. The majority of the selected neurons in the deep tissue were from the hippocampus, which was deeper than 1-mm. The fluorescence signal (ΔF/F_0_) of each selected neuron was plotted in Figure 5d,f. In the superficial cortex, the neuronal activities were rapidly suppressed after isoflurane inhalation and recovered when the anesthesia stopped. In the hippocampus, the majority of neurons showed the same performance as the cortical neurons, while the fluorescent signal of some neurons (Nos. 5 and 6) was increased dramatically, which was different from the response observed for the cortical neurons. Other neurons (Nos. 9 and 10) were still suppressed after isoflurane inhalation was stopped. One possible reason is that different cell types have different responses to isoflurane, which has been observed in in vitro experiments [29].

The neuronal activity in the mouse brains was recorded following anesthetization by propofol injection (Fresenius Kabi Austria GmbH company). We took 8000 time-lapse images with a 10 Hz frame-rate (800 s in total acquisition time) to record the neuronal activity in mice. After 120 s at an oxygen flow rate of 2 L/min, 26mg/kg propofol was injected through the caudal vein. In Figure 6a, we show the standard deviation projection of the first 1200 time-lapse images (120 s); two subregions of the superficial cortex and longitudinal plane were zoomed in on, as shown in Figure 6c,e. In each subregion, we randomly selected 10 neurons and plotted the fluorescence signal fluctuation (ΔF/F_0_), as shown in Figure 6d,f. Based on the fluorescence curves, we found that neuronal activity in both the superficial cortex and the hippocampus was rapidly suppressed when propofol was injected (Figure 6b). In the recovery process, the majority of neurons in the superficial cortex recovered after 400 s, and we also found that neuron No.4 quickly recovered from anesthesia at 200 s. In the deep brain region, the majority of the neurons were still suppressed until 800 s, while three neurons (Nos. 2, 4, 5) exhibited neuronal activity within 800 s.

## 4. Discussion and Conclusions

In our experiment, we observed single-cellular neural dynamics using different anesthetics. Calcium activity recovered rapidly after the mice stopped inhaling isoflurane, while calcium activity after propofol injection was continuously inhibited for at least 300 s. One possible explanation for this phenomenon is the difference in the pharmacokinetics of the two anesthetics. Isoflurane has high-fat solubility, which enables easy entry to brain tissue through the blood-brain barrier, and is mainly excreted from the body through the respiratory tract in its original form [30]. On the other hand, propofol is mainly metabolized by the liver, and its metabolism correlates with the volume of distribution and blood clearance [31]. Previous studies have also shown that anesthetics have different affinities for different cell types, which is a possible reason for the heterogeneity in the cellular dynamics. We intend to study the calcium signal activity of different types of neurons in response to anesthetics in future work.

In summary, we present a wide-field mesoscopic technique that simultaneously images the superficial cortex and deep brain regions by implanting chronic microprism windows. This method facilitated high-quality, in vivo imaging of the superficial cortex and an additional longitudinal view extending to 1.5-mm depth. The experimental results show that, in vivo, this modality achieved large-scale, high spatio-temporal resolution structural imaging of microglia in mouse brains. Furthermore, we applied this method to explore the effects of isoflurane and propofol, and observed the neuronal dynamics during the process of anesthesia and recovery. The results demonstrated different anesthesia effects of inhaling isoflurane and injecting propofol on neuronal activity in the mouse cortex, indicating different pathways of anesthesia.

## Figures and Tables

**Figure 1 biosensors-12-00567-f001:**
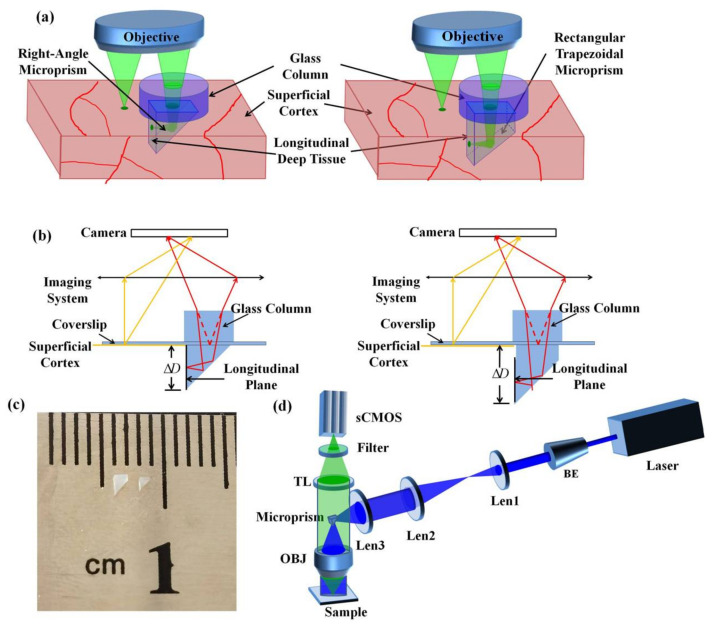
Schematic diagram of the multi-plane imaging technique. (**a**) Schematic of the experimental setup showing the microprism implant. (**b**) Schematic diagram of simultaneous imaging of side and top views. (**c**) Rectangular trapezoidal microprism (left) and right-angle microprism (right). (**d**) The scheme of the customized-built wide-field mesoscope, with brain-wide FOV and cellular resolution. BE: Beam expander; OBJ: Objective; TL: Tube Lens.

**Figure 2 biosensors-12-00567-f002:**
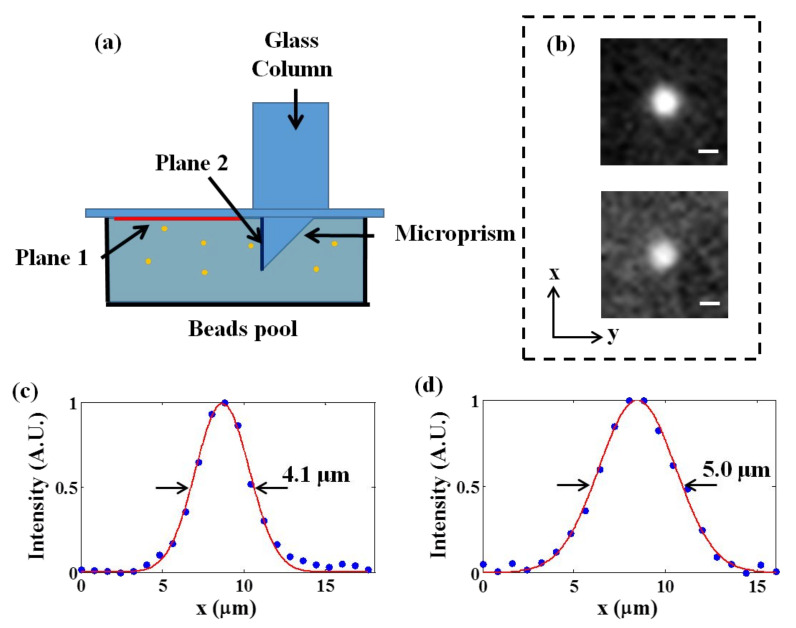
Characterization of the imaging system. (**a**) The scheme of imaging fluorescent microspheres. (**b**) The experimental PSFs in the horizontal plane of Plane 1 (upper) and Plane 2 (lower). Scale bar: 3 μm. (**c**,**d**) Lateral intensity profiles of the fluorescent microspheres on Plane 1 and Plane 2, respectively.

**Figure 3 biosensors-12-00567-f003:**
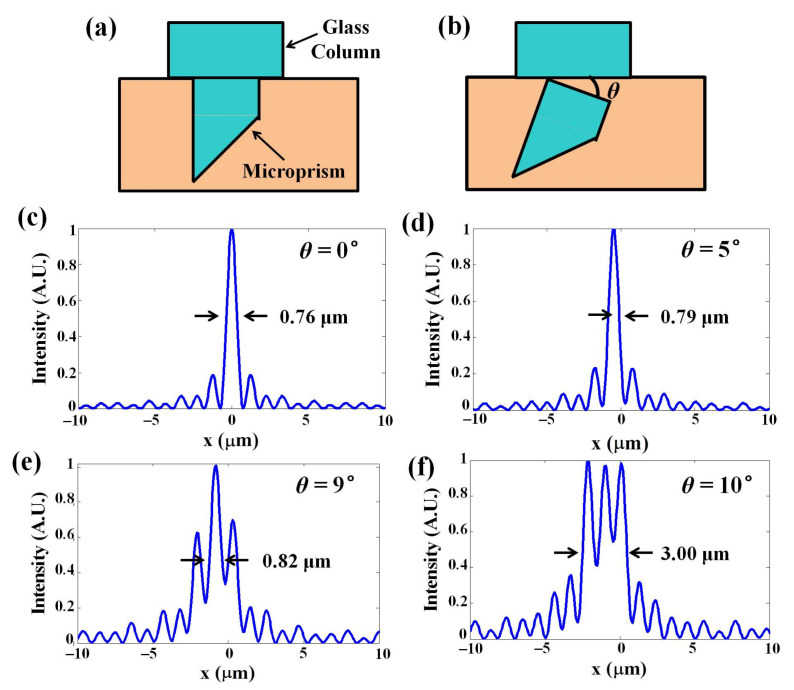
Simulations of microprism tilting. (**a**,**b**) Simulated intensity distributions with various tilt angles between the x axis and the glass surface, in the case of 0.5 NA and a 3-mm-thick glass at (**c**) 0°, (**d**) 5°, (**e**) 9° and (**f**) 10°.

**Figure 4 biosensors-12-00567-f004:**
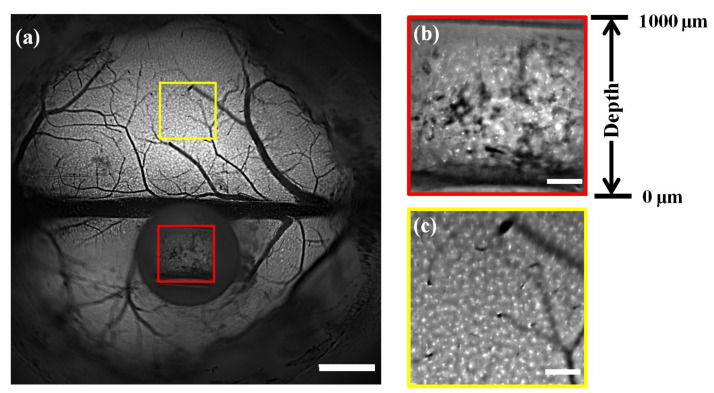
In vivo C×3Cr1-GFP mouse brain imaging. (**a**) Large-FOV image of Cx3Cr1-GFP mouse brains using a custom-built wide-field fluorescence mesoscope. Scale bar: 1 mm. (**b**,**c**) Magnified views of the subregions in the longitudinal plane and superficial cortex marked in (**a**). Scale bar: 200 μm.

**Figure 5 biosensors-12-00567-f005:**
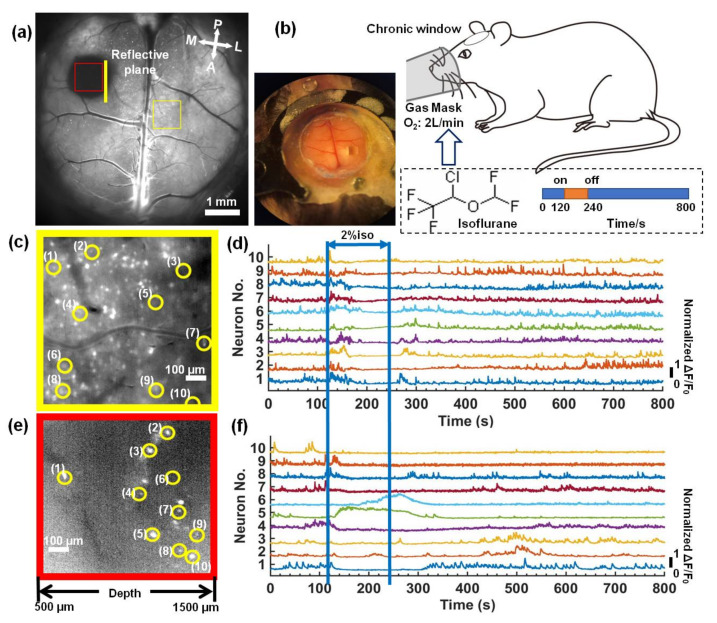
Deep in vivo functional imaging of mouse brains anesthetized by isoflurane. (**a**) The standard deviation projection of 600 time-lapse images of the mouse brains. Scale bar: 1 mm. (**b**) A schematic diagram of the experiment. (**c**,**e**) Magnified views of the subregions marked in (**a**). Scale bar: 100 μm. (**d**,**f**) Fluorescence signal fluctuation (ΔF/F_0_) for neurons in yellow circles in (**c**,**e**).

**Figure 6 biosensors-12-00567-f006:**
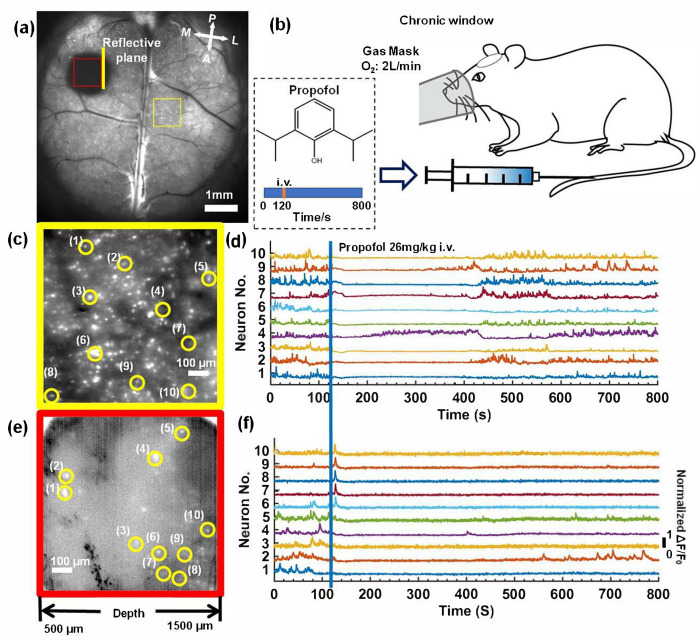
Deep in vivo functional imaging of mouse brains anesthetized by propofol. (**a**) The standard deviation projection of 600 time-lapse images of the mouse brains. Scale bar: 1 mm. (**b**) A schematic diagram of the experiment. (**c**,**e**) Magnified views of the subregions marked in (**a**). Scale bar: 100 μm. (**d**,**f**) Fluorescence signal fluctuation (ΔF/F_0_) for neurons in yellow circles in (**c**,**e**).

## Data Availability

The data presented in this study are available on request from the corresponding author.

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
