# Peer review of "Simultaneous Observation of Mouse Cortical and Hippocampal Neural Dynamics under Anesthesia through a Cranial Microprism Window"

_biosensors, 2022, doi:10.3390/bios12080567_

Round 1

Reviewer 1 Report

In this work, the authors combined an extended-field-of-view microscopy with a novel prism-based cranial windows to provide a longitudinal view, which carried out the simultaneous imaging of deep- brain regions and superficial cortex. And they designed a new rectangular-trapezoidal microprism cranial window to increase the depth of observation from 1 mm to 1.5mm, and meanwhile, reducing the brain injury. In the experiment, they first validated the method by the structural imaging of microglia cells in the superficial cortex and deep-brain regions. And the new method was further used to study neuronal activity from the mouse brains in the states under awake or anesthesia. It is interesting and show great potential for the advanced optical imaging in biomedical fields. I think this work can be accepted by the present journal.

Author Response

We thank you sincerely for reviewing our draft.

Reviewer 2 Report

In this work, the authors combine chronic micro prism windows and wide-field microscopy to image the whole superficial cortex and deeper. 

Briefly, the authors inserted the micro prism window to increase imaging depth to 1mm and uses a multi-planar imaging approach to correct for the depth difference between the surface cortex and deep tissues, allowing simultaneous imaging of both. The authors also improved their method by using a rectangular trapezoidal micro prism to improve imaging depth to 1.5mm. The results show microglia in the superficial cortex and deep tissues. During anesthesia and recovery with isoflurane and propofol, authors record dynamic neuronal activity in the surface cortex and at different depths.

Overall, the work is straightforward and shows an implementation of an existing technique and the authors' efforts to improve the current method. The novelty is progressive and the impetus is on the evaluation of the method rather than method innovation. That is my main criticism. A few other comments are listed below.

Line 103: Please verify the model for the camera: The Flash 4.0 model is not an sCMOS as far as I know.

Line 160: Please add the time (imaging time or total time) for 1200 time-lapse

Figure 1: Why is the micro prism placed between the objective lens and the camera? This schematic is confusing.

Section 4: The discussion illustrates simulation results (Fig. 6). Should this not be in the results section?

Figures 5 and 4 show a similar mouse cartoon with an added intravenous syringe. It would be easier if authors described what it is or used colors in the comic to focus readers on what is essential.

Section 2: The methods section for beads, propofol, and isoflurane treatment is missing in the methods section but mixed in with the results in lines 124, 150, and 176. It would be more organized if the methods appeared in the method section instead of the results. Also, vendor details of the beads and some other reagents are missing, which are critical for reproducing these results.

All the best

Author Response

In this work, the authors combine chronic micro prism windows and wide-field microscopy to image the whole superficial cortex and deeper.

Briefly, the authors inserted the micro prism window to increase imaging depth to 1mm and uses a multi-planar imaging approach to correct for the depth difference between the surface cortex and deep tissues, allowing simultaneous imaging of both. The authors also improved their method by using a rectangular trapezoidal micro prism to improve imaging depth to 1.5mm. The results show microglia in the superficial cortex and deep tissues. During anesthesia and recovery with isoflurane and propofol, authors record dynamic neuronal activity in the surface cortex and at different depths.

  1. Overall, the work is straightforward and shows an implementation of an existing technique and the authors' efforts to improve the current method. The novelty is progressive and the impetus is on the evaluation of the method rather than method innovation. That is my main criticism. A few other comments are listed below.

Thank you for pointing out this issue. Simultaneous imaging of the cortex is crucial for biological studies, such as the formation of memory. The current study has revealed that memory is formed in the hippocampus and distributed and stored in the cortex. But current technologies can only image small portions of the cortex or image the deep hippocampus. To the best of our knowledge, our technique is the first for simultaneous imaging the hippocampus and the cortex at centimeter-scale, cellular resolution, which provided a necessary tool for various biological studies.

  1. Line 103: Please verify the model for the camera: The Flash 4.0 model is not an sCMOS as far as I know.

Thanks for pointing out this issue. We have updated the model of sCMOS camera “Flash 4.0” with its full model “ORCA-Flash4.0 V3”, as shown in http://www.hamamatsu.com.cn/product/16669.html

Revisions in Line 108:

“The fluorescence is collected by an epi-fluorescence setup including the same objective, the tube lens (MVPLAPO 1X, Olympus), a filter (wavelength: 520 ± 12.5 nm/630 ± 46.5 nm 87-241, Edmund), and the sCMOS (ORCA-Flash4.0 V3, HAMAMATSU).”

  1. Line 160: Please add the time (imaging time or total time) for 1200 time-lapse

Thanks for the suggestion. The exposure time of the camera for a single image is 100 ms. For each experiment, we took 8000 time-lapse images in 800 seconds. And we also applied the first 1200-frame time-lapse images to show the standard deviation projection.

For better understanding for readers, we revised the draft.

Revisions in Line 198 and 219

“We took 8000 time-lapse images with 10 Hz frame-rate (800 seconds in total acquisition time) to record the neuronal activity in mice. The first 1200 time-lapse images (120 seconds) in awaken mice brains were applied to show the standard deviation projection in Figure 5(a).”

“We took 8000 time-lapse images with 10 Hz frame-rate (800 seconds in total acquisition time) to record the neuronal activity in mice……In Figure 6(a), we showed the standard deviation projection of the first 1200 time-lapse images (120 seconds)”

  1. Figure 1: Why is the micro prism placed between the objective lens and the camera? This schematic is confusing.

Thanks for this suggestion. The microprism placed between the objective lens and the camera plays a similar role as a dichroic mirror, which can separate the coherent excitation laser and the incoherent emission fluorescence, as previously described in Adam Cohen (Biomed Opt Express 8(12), 5794-5813, 2017) and our previous work (Biomed Opt Express 12(4), 1858, 2021). The coherent laser can be focused in a very small region and reflected in the sample, while the incoherent fluorescence is distributed on the aperture plane of the objective, so only a small portion of fluorescence is blocked and most of them can be collected by the camera.

For better understanding for readers, we revised the draft.

Revisions in Line 104:

“The microprism placed between the objective lens and the camera plays a similar role as a dichroic mirror, which can separate the coherent excitation laser and the incoherent emission fluorescence [13].”

  1. Section 4: The discussion illustrates simulation results (Fig. 6). Should this not be in the results section?

Thanks for your constructive suggestion. We have moved the simulation results (Fig. 6) to the results section in Section 3.1 in our revised draft, shown in Line155-170.

  1. Figures 5 and 4 show a similar mouse cartoon with an added intravenous syringe. It would be easier if authors described what it is or used colors in the comic to focus readers on what is essential.

Thanks for your constructive suggestion. In order to distinguish the difference between Figures 4 (b) and 5 (b), we have added the different experimental settings and created two insets to emphasize them.

Revisions: Enlarging intravenous syringe image, adding color and an arrow to let the readers focus on the difference. The revised Figures 5 and 6.               

  1. Section 2: The methods section for beads, propofol, and isoflurane treatment is missing in the methods section but mixed in with the results in lines 124, 150, and 176. It would be more organized if the methods appeared in the method section instead of the results. Also, vendor details of the beads and some other reagents are missing, which are critical for reproducing these results.

Thanks for pointing out this issue. We have reorganized the section and added Section 2.4 for describing the treatment of the beads, propofol, and isoflurane. Besides, we also add the vendor details of the beads “T14792, Thermo Fisher Scientific” in Line 127.

Revisions in Line 127-139:

“For measuring the resolution of our system, the 3.1-mm-diameter green fluorescent beads (T14792, Thermo Fisher Scientific) were fixed in a pool with 1% low-gelling temperature agarose (A9414-25G, Sigma), and then a microprism was inserted in the pool and covered it with a coverslip. In our in vivo mouse experiments, we applied the two different anesthesia treatments to the mice. For isoflurane inhalation, the mouse were fixed under the microscope and inhaled oxygen through a mask connected to an anesthesia machine (R530IE, RWD life science company, China) at a flow rate of 2 L/min. At 120 s after the beginning of the experiment, we delivered 2% isoflurane (RWD life science company, China) at the oxygen delivery rate of 2 L/min. At 240 s, we stopped the isoflurane inhalation and waited for the recovery of the mouse; For propofol injection (Fresenius Kabi Austria GmbH company), we also placed the mouse under the microscope and inhaled oxygen to the mouse. At 120 s after the beginning of the experiment, we injected propofol through the tail vein (26mg/kg).”

Reviewer 3 Report

This article, which achieved the simultaneous imaging of deep-brain regions and superficial cortex by a microscopy with a cranial microprism window, is of interest to the wide readers and well presented. It is recommended to modify before publication. The article has the following questions: 

1.     Please clarify the different observational abilities (depth, resolution, etc) when using different microprisms. And why were they designed different heights, 1mm and 1.5 mm? In Figure 1b, only the right-angle microprism was shown. The rectangular trapezoidal microprism should be added.

2.     Figure 2: Does the resolution change with depth in Plane 2?

3.     Figure 4f: No 9 and 10 seemed suppressed during the observation. Did they recover but were too deep to detect, or did they not recover? Is there any relevant literature supporting the recover ability?

4.     The isoflurane and the propofol showed different function in anaesthesia. Please add some discussion about it.

Author Response

This article, which achieved the simultaneous imaging of deep-brain regions and superficial cortex by a microscopy with a cranial microprism window, is of interest to the wide readers and well presented. It is recommended to modify before publication. The article has the following questions:

  1. Please clarify the different observational abilities (depth, resolution, etc) when using different microprisms. And why were they designed different heights, 1mm and 1.5 mm? In Figure 1b, only the right-angle microprism was shown. The rectangular trapezoidal microprism should be added.

Thanks for pointing out this issue. The right-angle microprism (1 mm × 1 mm × 1 mm) provides 0-1 mm depth range, and the rectangular trapezoidal microprism (1 mm × 1 mm × 1.5 mm) provides the imaging depth from 0.5-1.5mm. We emphasize this issue in the revised draft in Line 83.

Revision:

“The right-angle microprism provides the imaging depth from 0 to 1mm, and the rectangular trapezoidal microprism provides the imaging depth from 0.5 to 1.5mm.”

And resolution measurement of the right-angle microprism is shown in Section 3.1, and we apply the 3.1-mm-diameter fluorescent microspheres right-angle microprism for the experiment. The results show that the resolutions of the surface and longitudinal planes are 4.1 μm and 5.0 μm, respectively. The rectangular trapezoidal microprism has more imaging depth and further affects the resolution, but in our previous study (Biomed Opt Express 12(4), 1858, 2021), the resolution decreased by less than 0.5 μm, which does not affect to observe of the neurons.  

The right-angle microprism (1 mm × 1 mm × 1 mm) is a widely-used optical element in deep-tissue imaging, as shown in the previous article, but simultaneous imaging of deep nuclei and the cortex keeps challenging. To the best of our knowledge, we achieved simultaneous imaging of the superficial plane and deep cortex for the first time, by combining the microprism and multi-planar method. But the right-angle microprism (1 mm × 1 mm × 1 mm) provides only 1 mm imaging depth, hardly observing the hippocampus. To further improve the depth, increasing the size of the right-angle microprism is an available method, such as the 1.5 mm × 1.5 mm × 1.5 mm microprism, but it increases brain damage. Thus, we propose another method: rectangular trapezoidal microprism (1 mm × 1 mm × 1.5 mm) to improve the depth to 1.5 mm with less brain damage. We emphasize this issue in Line 70:

“To further improve the imaging depth to 1.5mm with minimized brain damage, we designed a rectangular trapezoidal microprism, providing the imaging depth from 0.5 to 1.5mm.”

We added rectangular trapezoidal microprism method diagram in Figure 1.

  1. Figure 2: Does the resolution change with depth in Plane 2?

Resolution in Plane 2 does not change with depth. As plane 2 is reflected by microprism, all points in different depths in Plane 2 are mapped into the focal plane and have an equivalent optical path, so they have no difference in resolution. We will clarify this issue in our revised draft in Line 152.

Revisions:

“Because Plane 2 is reflected by microprism to the camera, all points in different depth in Plane 2 are mapped into the focal plane and have an equivalent optical path, so they should have no difference in resolution”

  1. Figure 4f: No 9 and 10 seemed suppressed during the observation. Did they recover but were too deep to detect, or did they not recover? Is there any relevant literature supporting the recover ability?

Thank you very much for your question.

In our experiments, we observed that different neurons have different calcium activities for recovery from anesthesia. During the observation process, the observation depth did not change, so the difference in neuronal calcium events may be due to the difference in the neuron's own response to anesthetics. In in vitro experiments, Spiegel Iris A et al. found that isoflurane inhibits presynaptic Ca transients by activating glutamatergic neurons and somatostatin-GABAergic interneurones. [Speigel Iris A,Patel Kishan,Hemmings Hugh C,Distinct effects of volatile and intravenous anaesthetics on presynaptic calcium dynamics in mouse hippocampal GABAergic neurones.[J] .Br J Anaesth, 2022, 128: 1019-1028.] Owing to we used nonspecific GCaMP6f to label neurons; we cannot determine whether this difference is due to differences in neuron type. This will be the focus and direction of our follow-up research.

We will clarify this issue in our revised draft in Line 214.

Revision:

“Other neurons (Nos. 9 and 10) were still suppressed after the isoflurane inhaling was stopped. One possible reason is that different cells types have different response to the isoflurane, which has been revealed in in vitro experiments [29].”

  1. The isoflurane and the propofol showed different function in anesthesia. Please add some discussion about it.

Thank you very much for your suggestion, we have added the following to the article.

In our experiment, we observed the single-cellular neural dynamics under different anesthetics. The calcium events recovered rapidly after the mice stopped inhaling isoflurane, while the calcium event after propofol injection was continuously inhibited for at least 300 seconds. One possible explanation for this phenomenon is the difference in the pharmacokinetics of the two anesthetics. Isoflurane has high-fat solubility, which can easily enter the brain tissue through the blood-brain barrier and is mainly excreted from the body through the respiratory tract in its original form. (Klaus Otto, Anne-Kathrin von Thaden, The Laboratory Mouse (Second Edition), 2012). On the other hand, propofol is mainly metabolized by the liver, and its metabolism correlates with the volume of distribution and blood clearance. (Anushirvan minokadeh, William Wilson, Cardiac Intensive Care (Second Edition), 2010) Further, previous studies have also shown that anesthetics have different affinities on different cell types, which is a possible reason for the heterogeneity in the cellular dynamics. So we will study the calcium signal activity of different types of neurons in anesthetics in our future work.